# The Fall Armyworm, *Spodoptera frugiperda* (Lepidoptera: Noctuidae), Influences *Nilaparvata lugens* Population Growth Directly, by Preying on Its Eggs, and Indirectly, by Inducing Defenses in Rice

**DOI:** 10.3390/ijms24108754

**Published:** 2023-05-15

**Authors:** Chen Qiu, Jiamei Zeng, Yingying Tang, Qing Gao, Wenhan Xiao, Yonggen Lou

**Affiliations:** 1State Key Laboratory of Rice Biology & Ministry of Agriculture Key Laboratory of Agricultural Entomology, Key Laboratory of Biology of Crop Pathogens and Insects of Zhejiang Province, Institute of Insect Sciences, Zhejiang University, Hangzhou 310058, China; cqiu2019@163.com (C.Q.); tsengchiamei@zju.edu.cn (J.Z.); tyy034899@zju.edu.cn (Y.T.); 11916008@zju.edu.cn (Q.G.); whxiao@zju.edu.cn (W.X.); 2Hainan Institute, Zhejiang University, Sanya 572025, China

**Keywords:** herbivore-induced plant defense, signaling pathway, intraguild predation, jasmonoyl-isoleucine, abscisic acid

## Abstract

The fall armyworm (FAW), *Spodoptera frugiperda*, has become one of the most important pests on corn in China since it invaded in 2019. Although FAW has not been reported to cause widespread damage to rice plants in China, it has been sporadically found feeding in the field. If FAW infests rice in China, the fitness of other insect pests on rice may be influenced. However, how FAW and other insect pests on rice interact remains unknown. In this study, we found that the infestation of FAW larvae on rice plants prolonged the developmental duration of the brown planthopper (BPH, *Nilaparvata lugens* (Stål)) eggs and plants damaged by gravid BPH females did not induce defenses that influenced the growth of FAW larvae. Moreover, co-infestation by FAW larvae on rice plants did not influence the attractiveness of volatiles emitted from BPH-infested plants to *Anagrus nilaparvatae*, an egg parasitoid of rice planthoppers. FAW larvae were able to prey on BPH eggs laid on rice plants and grew faster compared to those larvae that lacked available eggs. Studies revealed that the delay in the development of BPH eggs on FAW-infested plants was probably related to the increase in levels of jasmonoyl-isoleucine, abscisic acid and the defensive compounds in the rice leaf sheaths on which BPH eggs were laid. These findings indicate that, if FAW invades rice plants in China, the population density of BPH may be decreased by intraguild predation and induced plant defenses, whereas the population density of FAW may be increased.

## 1. Introduction

Upon infestation by herbivores, plants recognize herbivore-associated molecular patterns and then initiate early signaling events; these include an increase in levels of cytosolic Ca^2+^ [1], a burst of reactive oxygen species (ROS) [2] and the activation of mitogen-activated protein kinases (MAPKs) [3]. Early signaling events thus activate signaling pathways mediated by phytohormones, such as jasmonic acid (JA), salicylic acid (SA), ethylene (ET) and abscisic acid (ABA); the activated phytohormone signaling pathways regulate the expression of defense-related genes and the biosynthesis of defense-related compounds, which in turn increase the resistance of plants to herbivores [4]. These herbivore-induced defense responses may affect the fitness of both conspecific and non-conspecific herbivores, and of other organisms sharing the same host plant [5,6,7].

Rice is one of the most important food crops in the world. It suffers from infestations by many insect pests, including the brown planthopper (BPH) *Nilaparvata lugens* (Stål) (Hemiptera: Delphacidae). In China, BPH is a destructive insect pest on rice that generally results in yield losses of up to 30% if not controlled, and mainly immigrates from the north of Vietnam [8,9]. BPH damages rice plants by feeding on phloem sap, laying eggs in tissues and/or transmitting viruses [10,11,12,13]. It has been well documented that the infestation of BPH gravid females alters the biosynthesis of many defensive signals, such as JA, jasmonoyl-isoleucine (JA-Ile), SA, ET and H_2_O_2_; these signal-mediated pathways then enhance both the expression of defensive genes and the production of defensive compounds, such as volatiles, phenolamides and the activity of trypsin protease inhibitors (TrypPIs) [14,15,16]. Phenolamides and TrypPIs are reported to be defensive compounds that act directly against BPH [17]; moreover, volatiles emitted from rice plants infested by BPH are attractive to *Anagrus nilarparvatae* (Pang et Wang) (Hymenoptera: Mymaridae), the egg parasitoid of BPH [18]. Hence, BPH infestation enhances the direct and indirect resistance of rice to BPH.

The fall armyworm (FAW), *Spodoptera frugiperda* (J.E. Smith) (Lepidoptera: Noctuidae), is a destructive agricultural pest all over the world. In Brazil, for example, farmers need take about USD 600 million a year to control it [19]. FAW larvae feed on at least 353 plant species, including maize (*Zea mays* L.) and rice (*Oryza sativa* L.) [20]. Young FAW larvae generally aggregate to feed on leaves of host plants, whereas 3rd-instar and above FAW larvae disperse and consume leaves [21,22]. FAW originated in the tropical-subtropical part of the Western Hemisphere [23]. Since 2016, it has successfully invaded west and central Africa, damaging corn production [19,24]. In July 2018, FAW arrived in Yemen and India [25], and in January 2019, it was found in Yunnan province, China [26]. Nowadays, it has become one of the most important pests on corn in China [27]. FAW has two host plant-related strains, corn and rice [28]. The former strain primarily feeds on corn, cotton and sorghum, whereas the latter prefers rice and several pasture grasses. Despite their morphological similarity, these two strains exhibit distinct mating behavior, pheromone composition and developmental patterns [28]. Although the genetic background of its populations in China is corn [29], FAW has been sporadically found feeding on rice in the field in Guangxi, Fujian and Hubei provinces [27,30,31]. Moreover, because it can continue to reproduce on rice plants in the laboratory [30], FAW may well infest rice in China. FAW larvae infestation can induce defense responses, changing the physiological and biochemical status of plants [32,33,34], which may in turn influence the fitness of other insect pests on rice. In addition, besides cannibalizing its own species, FAW preys on other herbivores, such as *Helicoverpa zea* (Boddie) (Lepidoptera: Noctuidae) [35], *Busseola fusca* (Fuller) (Lepidoptera: Noctuidae) [36], *Sesamia calamistis* (Hampson) (Lepidoptera: Noctuidae) [37] and *Chilo partellus* (Swinhoe) (Lepidoptera: Crambidae) [36], a phenomenon known as intraguild predation [38,39]. However, whether and how current rice insect pests, such as BPH, and the potentially invasive leaf-chewer FAW interact, and how that interaction subsequently influences the interacting parties, remains unknown.

To address the above issues, in this study, we investigated the interaction between FAW and BPH. We found that FAW larvae not only prey on BPH eggs but also prolong the developmental duration of BPH eggs by triggering a systemic increase in levels of JA-Ile, ABA and defensive compounds in the rice leaf sheaths on which BPH eggs are laid. In addition, compared to those that do not prey on BPH eggs, FAW larvae preying on BPH eggs grow faster. The results demonstrate that, if FAW infests rice, FAW can benefit by preying on BPH eggs, whereas the BPH population might be inhibited because its eggs will be predated and its eggs’ development will be prolonged.

## 2. Results

### 2.1. Effects of FAW Larvae-Infested Rice Plants on BPH Performance

We investigated the influence of plants with different damage levels inflicted by FAW larvae on the feeding, development, survival and fecundity of BPH. No difference was observed in the amount of honeydew excreted by newly emerged BPH female adults for 1 day, an indicator of the amount of food intake, between plants that were pre-infested by a 3rd-instar FAW larva for 1–3 days and non-infested plants (Appendix A; Appendix A). The number of eggs laid by BPH female adults for 10 days on plants that were individually pre-infested by a 3rd-instar FAW larva for 1 or 4 days was similar to the number of eggs laid by BPH female adults for 10 days on non-infested plants (Appendix A; Appendix A). Pre-infestation of 3rd-instar FAW larvae (each plant had one larva) for 5 days, which resulted in the maximum damage to leaves (almost all of the leaves were eaten) (Appendix A; Appendix A), did not influence the developmental duration of BPH nymphs (Appendix A; Appendix A), the number of BPH eggs laid by BPH female adults for 24 h or the hatching rate and developmental duration of BPH eggs (Appendix A; Appendix A).

We also measured the post-infestation of FAW 1st- and 2nd-instar larvae for 7 days on the survival and development of BPH eggs. Neither the number of BPH eggs laid by BPH female adults for 24 h, nor the hatching rate and developmental duration of BPH eggs were influenced by the post-infestation of 2 or 3 1st-instar FAW larvae per plant for 7 days (Figure 1a–f; Appendix A). However, when plants were individually post-infested by 3 2nd-instar FAW larvae for 7 days, the developmental duration of BPH eggs in these plants was longer by 1 day than that of BPH eggs on non-infested plants; however, the hatching rate of BPH eggs was not affected (Figure 1a–f; Appendix A).

### 2.2. Effects of BPH-Infested Rice Plants on Performance of FAW Larvae

When newly hatched FAW larvae had fed for 8 to 9 days on the non-BPH infested parts of plants (meaning FAW larvae could not reach BPH eggs) that were pre-infested by 15 gravid BPH females for 2 or 5 days (whether or not BPHs were later removed), their larval mass was similar to that of FAW fed on control plants (without BPH infestation) (Appendix A; Appendix A). However, when newly hatched FAW larvae were allowed to feed on entire BPH-infested plants (meaning FAW could reach BPH eggs), they grew faster than those fed on control plants: by day 5.5, the mass of FAW larvae fed on BPH-infested plants was 1.57-fold higher than the mass of FAW larvae fed on control plants (Figure 2a; Appendix A). To examine whether the fast growth of FAW larvae on BPH-infested plants was due to the presence of BPH eggs, we allowed 3rd-instar FAW larvae to feed on plants infested by gravid BPH females (containing BPH eggs). Our results showed that these larvae grew faster than those fed on non-infested plants; in addition, BPH eggs on BPH-infested plants were predated by FAW larvae (Figure 2b,c; Appendix A). Moreover, by days 2 and 3, the mass of FAW larvae fed on rice leaf sheaths together with 400 BPH eggs per day was significantly higher than the mass of those fed on rice leaf sheaths alone (Figure 2d; Appendix A). These data demonstrate that BPH infestation on rice can facilitate the growth of FAW larvae by providing eggs for FAW larvae, whereas the defense responses in rice induced by BPH infestation have no effect on the growth of FAW larvae.

### 2.3. FAW Larvae Infestation Does Not Influence the Attractiveness of BPH-Infested Plants to the Parasitoid A. nilaparvatae

When given a choice between BPH eggs on BPH-infested plants and BPH eggs on BPH+FAW-infested plants, the parasitoid *A. nilaparvatae* showed no preference for BPH eggs: the parasitism rate by the parasitoid of BPH eggs on plants that were infested by 10 gravid BPH females for 24 h was similar to the parasitism rate of BPH eggs on plants that were infested by 10 gravid BPH females together with 1 4th-instar larva or 2 or 3 3rd-instar FAW larvae for 24 h (Appendix A; Appendix A). No difference in the number of BPH eggs was observed between pairs of plants (Appendix A).

### 2.4. FAW Larvae Infestation Enhances BPH-Induced Levels of JA-Ile and ABA but Not Levels of JA, SA and H_2_O_2_ in Rice

We investigated the change in levels of BPH-induced JA, JA-Ile, SA, ABA and H_2_O_2_ in the leaf sheaths (the feeding and oviposition position of BPH) of rice plants after they were infested by FAW larvae. In summary, the levels of JA-Ile and ABA in the leaf sheaths of plants that were infested by 10 gravid BPH females for 24 h, followed by the infestation of 3 2nd-instar FAW larvae for 3 days, was 3-fold and 2.5-fold, respectively (higher than that in the leaf sheaths of plants that were infested by 10 gravid BPH females for 24 h alone), whereas no difference was observed in the levels of JA, SA and H_2_O_2_ between the 2 plant treatments (Figure 3a–e; Appendix A).

### 2.5. FAW Infestation Influences BPH-Induced Levels of Phenolamides and Flavonoids in Rice

Levels of 3 flavonoids—astragalin (Figure 4c), luteolin 7-*O*-glucoside (Figure 4d) and schaftoside + isoschaftoside (Figure 4e)—in plants that were infested by gravid BPH females for 1 day, followed by infestation by FAW larvae for 1 day, was significantly higher than levels in plants that were infested only by gravid BPH females for 1 day (Appendix A). Interestingly, FAW larvae infestation reduced levels of feruloylputrescine (Figure 4a; 1 day after FAW larvae infestation) and luteolin (Figure 4b; 1 and 7 days after FAW larvae infestation) in BPH-infested plants (Appendix A). No difference was observed in levels of phenolamides and flavonoids at other time points between plants jointly infested by gravid BPH females and FAW larvae, and plants infested by gravid BPH females alone (Appendix A).

## 3. Discussion

In the study, we did not find that rice plants infested by BPH influenced the growth of FAW larvae (Appendix A). However, preying on BPH eggs on rice plants did accelerate the growth of FAW larvae (Figure 2). On the other hand, we found that post-infestation of FAW larvae (three 2nd-instar larvae) for seven days prolonged the developmental duration of BPH eggs, although rice plants with other damage levels inflicted by FAW larvae had no effect on the performance of BPH (Figure 1 and Appendix A). These findings demonstrate that the rice plant-mediated interaction between FAW and BPH was asymmetric: plants infested by FAW larvae influence the performance of BPH, but not vice versa. Moreover, FAW larvae could suppress the growth of the BPH population by preying on BPH eggs and prolonging the development of BPH eggs, and FAW could benefit from this interaction.

It has been well documented that chewing herbivores induce stronger defenses in plants than do sucking herbivores as the former generally cause more damage to plants than the latter do [39,40,41,42,43,44]. For instance, infestation by larvae of the rice striped stem borer (*Chilo suppressalis* (Walker) (Lepidoptera: Pyralidae), SSB), which bore into rice leaf sheaths or stems and feed inside these [40], induces a higher JA level than does BPH infestation [44]. Similarly, the expression level of *LOX1*, which encodes lipoxygenase (LOX) 1, a key enzyme in the biosynthesis of JA, in Arabidopsis infested by the green peach aphid (*Myzus persicae* (Sulzer) (Hemiptera: Aphidoidea)) is significantly lower than that in Arabidopsis infested by chewing herbivores [43,45]. In this study, BPH prefers to stay at the base of rice plants and damages plants by feeding on the phloem sap and laying eggs in tissues, whereas FAW is a chewing herbivore and usually feeds on the leaves of rice plants. Thus, the different feeding styles of the two herbivores probably underlie the asymmetric plant-mediated interaction between BPH and FAW. However, it should be noted that BPH infestation activates multiple metabolic pathways, and that these changes strongly depend on the genotype involved [46]. Therefore, it is also possible that BPH infestation may influence the performance of FAW with the change in rice genotype.

FAW larvae are omnivorous, and intraguild predation brings nutritional and energy benefits, increasing the size, growth and development of individuals [47,48,49,50]. Hence, the fast growth rate of FAW feeding on BPH eggs is probably related to the high nutritional and energy benefits of the eggs. In general, the larval mass of herbivores is correlated with potential fecundity [51,52,53]. In this study, we observed that the mass of 5.5-day-old FAW larvae fed on entire BPH-infested plants (meaning FAW could reach BPH eggs) increased by 57% compared to the mass of 5.5-day-old FAW larvae fed on control plants. Thus, it could be expected that co-existence between BPH and FAW promotes an increase in the population density of FAW.

The developmental duration and the number of eggs laid by female adults of BPH are two main factors that influence the population dynamics of BPH [54]. For developmental duration, it has been reported that a one-day delay in the development of immature-stage BPH at the fourth generation decreases the density by 11.4% of the peak of the total BPH population at the fifth generation. Moreover, this one-day delay also reduces the population density of the parasitoid *A. nilaparvatae* by 13.6% [54]. For the number of eggs laid by BPH female adults, a 10% decrease at the fourth generation results in reductions (by 23.2% and 18.8%, respectively) in the density of the peak of the total BPH population at the fifth generation and in the density of the *A. nilaparvatae* population. Therefore, a one-day delay in BPH egg development and the decrease in the number of BPH eggs caused by FAW larval infestation will probably have a relatively large effect on the population dynamics of BPH, especially in places such as Hainan Island, China, where 10–12 BPH generations occur per year [55].

JA, SA, H_2_O_2_ and ABA signaling pathways play an important role in the herbivore-induced defense responses of plants, including rice [14,15,16]. Therefore, to explain why the infestation of FAW larvae on rice plants prolonged the developmental duration of BPH eggs, we investigated the change in levels of signaling molecules JA, JA-Ile, ABA, SA and H_2_O_2_ in plants when they were pre-infested by BPH for 24 h, followed by the infestation of FAW larvae or infestation by BPH alone. We found that levels of JA-Ile and ABA in the leaf sheaths of plants infested by BPH gravid females are enhanced (the sheaths are the main location of BPH feeding and oviposition) 3 days after infestation (Figure 3b,d), suggesting that the co-infestation of both BPH and FAW larvae activated JA- and ABA-mediated signaling pathways in rice more strongly than did the infestation of BPH alone. Both JA and ABA signaling pathways have been reported to play important roles in regulating BPH resistance in rice [14,15,16]. For example, Xu et al. (2021) reported that the hatching rates of BPH eggs were significantly higher for those laid on JA-deficient lines than for those laid on wild-type plants [56]. Zhou et al. [57] found that significantly fewer eggs were laid by BPH females on the mutant *osaba8ox3* (the knocked-down *OsABA8ox3* gene; *OsABA8ox3* is the key gene in ABA hydrolase genes) than on wild-type plants [57]. Moreover, the ABA signaling pathway has also been observed to reduce the hatching rate of BPH eggs [58]. Hence, the delay in the development of BPH eggs in rice plants that were infested by FAW larvae might be related to the activation of signaling pathways mediated by JA and ABA.

We also compared the difference in levels of phenolamides and flavonoids, two kinds of defensive compounds against herbivores in plants, including rice [59,60,61,62], using rice plants that were infested by BPH gravid females alone or co-infested by BPH gravid females and FAW larvae. Interestingly, FAW larvae infestation decreased levels of two compounds, feruloylputrescine and luteolin, in BPH-infested plants. Moreover, levels of three flavonoids—astragalin (Figure 4c), luteolin 7-*O*-glucoside (Figure 4d) and schaftoside + isoschaftoside (Figure 4e)—were higher in BPH-FAW-infested plants at one day after FAW infestation than in BPH-infested plants. Of these increased compounds, schaftoside and its isomers (isoschaftoside and neoschaftoside) have been reported to negatively influence BPH feeding [59,63]. Moreover, schaftoside can strongly bind with cyclin-dependent kinase 1 of BPH (*NlCDK1*) and inhibits the activation of *NlCDK1* as a kinase by suppressing phosphorylation on its Thr-14 site [64]. Therefore, it is possible that increased levels of these compounds, such as schaftoside + isoschaftoside, in rice delay the development of BPH eggs in rice tissues by permeating these eggs. Future investigations should determine the compounds that influence the development of BPH eggs.

Thus far, several studies have reported that non-host herbivore infestation could influence the attractiveness of volatiles emitted from host-infested plants to parasitoids (predators). For instance, *Cotesia marginiventris* (Cresson) (Hymenoptera: Braconidae), a parasitoid of *Spodoptera exigua* (Hübner) (Lepidoptera: Noctuidae), was more attracted to tomato plants infested with both the aphid *Macrosiphum euphorbiae* (Thomas) (Hemiptera: Aphididae) and its host than were plants infested with *S. exigua* alone [65]. The infestation of *Bemisia tabaci* (Gennadius) (Hemiptera: Aleyrodidae) decreases the attractiveness of spider mite-damaged plants to predatory mites [66]. More recently, Hu et al. (2020) reported that BPH females preferred to oviposit on plants that were infested by SSB larvae than on non-infested rice plants, as SSB larvae infestation results in an enemy-free space for BPH eggs: volatiles emitted from plants infested by SSB larvae and BPH gravid females were much less attractive to *A. nilaparvatae*, the egg parasitoid of BPH, than were volatiles emitted from plants infested by BPH gravid females alone [67]. However, here we did not find that FAW larvae infestation on plants influences the preference of the parasitoid *A. nilaparvatae* for BPH eggs (Appendix A), although there seem to be distinct differences in measurable volatiles between BPH-infested plants [18] and FAW-infested plants [68]. Similar results were also documented in other research systems. The infestation of *Euscelidius variegatus* (Kirschbaum) (Hemiptera: Cicadellidae), for example, did not alter the attractiveness of *Spodoptera littoralis* (Boisd) (Lepidoptera: Noctuidae)-infested maize plants to the parasitoid *C. marginiventris* [69]. These findings suggest that whether non-host herbivore infestation influences the attractiveness of host-damaged plants to parasitoids or predators depends on whether the volatile signals attractive to parasitoids or predators change.

In summary, we demonstrate that FAW larvae could facilitate their own growth by preying on BPH eggs on rice plants. Moreover, FAW larvae infestation could also prolong the development of BPH eggs by inducing the defense responses of rice. We propose that the delay in the development of BPH eggs in rice plants infested simultaneously by FAW larvae is probably due to the increase in signaling pathways mediated by JA and ABA and ensuing defensive compounds, such as flavonoids. These findings indicate that any infestation of rice plants in China by FAW—through intraguild predation and induced defensive responses—will probably decrease the population of BPH and change the composition of the arthropod community in rice. Moreover, because FAW benefits from its interaction with BPH, its population density and damage to plants may increase.

## 4. Materials and Methods

### 4.1. Plants and Insects

The seeds of the japonica rice variety, XiuShui 110 (XS110), were provided by China National Rice Research Institute, Hangzhou, China. Pre-germinated seeds of XS110 were cultured in a plastic bottle (diameter 8 cm, height 10 cm) in an illuminated incubator (28 ± 2 °C with a 14 h light period). Seven-day-old seedlings were grown in 25 L hydroponic boxes (length 51 cm, width 35 cm, height 17 cm) filled with a rice nutrient solution [70] and kept in a greenhouse (26 ± 2 °C 14 h light, 60% relative humidity). Twenty-five days later, seedlings were individually transferred to plastic pots (diameter 7 cm, height 9.5 cm) containing 350 mL of a nutrient solution. Plants were used for experiments 3–5 days after transplantation.

Colonies of BPH were originally obtained from rice fields in Hangzhou, China, and subsequently maintained on seedlings of the indica rice variety TN1 (susceptible to BPH) in a controlled climate room at 26 ± 2 °C, 12 h light phase and 80% relative humidity. Colonies of *A. nilaparvatae*, the egg parasitoid of BPH, were collected on rice plants containing BPH eggs from fields in Hangzhou, China, and propagated with BPH eggs on TN1 rice seedlings. FAW eggs were collected from maize fields in Hangzhou, China, and then larvae were individually reared on fresh maize leaves in Petri dishes (diameter 9 cm, height 1.5 cm).

### 4.2. Plant Treatment

For BPH infestation, plants were individually confined in glass cages (diameter 4 cm, height 8 cm, with 48 small holes, diameter 0.8 mm) into which 10 or 15 BPH gravid females were released (Appendix A). Plants with empty glass cages were used as controls. For FAW infestation, one or more FAW larvae were placed on the upper part of the plants, each of which was individually confined in a plastic cylinder (diameter 4 cm, height 50 cm; with a side ventilation opening (length 15.5 cm, width 7 cm) and a top opening, all of which were covered with 140-mesh nylon nets) (Appendix A). Plants confined in empty plastic cylinders were used as controls.

### 4.3. Bioassays

#### 4.3.1. Effects of FAW Larvae-Infested Rice Plants on BPH Performance

Because 3rd-instar and above larvae of FAW disperse when they consume leaves, we investigated the effect of plants with different damage levels inflicted by FAW 3rd-instar larvae on the feeding, fecundity, survival and development of BPH. To assess the impact of FAW larvae-infested plants on BPH feeding, a newly emerged female adult was put into a parafilm bag (3 × 4 cm; weighed in advance) attached to the leaf sheath (Appendix A) of a non-infested rice plant or a plant that was infested by a 3rd-instar FAW larva for 1, 2 or 3 days. Twenty-four hours later, the honeydew excreted by the BPH female adult was weighed by an analytical lab balance with a readability of 0.1 mg (to an accuracy of 0.1 mg). Each treatment was replicated 21~25 times.

To investigate the impact of FAW larvae-infested plants on the fecundity and oviposition behavior of BPH female adults, a non-infested plant or a plant that was infested by a 3rd-instar FAW larva for 1 or 4 days was confined in the glass cage into which a newly emerged BPH female and a male adult were introduced using the method described above. Ten days later, the number of eggs laid by BPH female adults on each plant was counted under a microscope. Each treatment was replicated 21~26 times.

To determine the impact of FAW larvae-infested plants on the survival of BPH nymphs, 15 newly emerged BPH nymphs were introduced on a non-infested plant or a plant that was infested by a 3rd-instar FAW larva for 5 days using the method described above (Appendix A). The number of BPH nymphs surviving on each plant was recorded every day until all nymphs had become adults. Each treatment was replicated 8~9 times.

To assess the impact of FAW larvae-infested plants on the hatching rate and developmental duration of BPH eggs, 10 gravid BPH females were allowed to lay eggs for 24 h on the leaf sheaths of a non-infested plant or a plant that was infested by a 3rd-instar FAW larva for 5 days using the method described above. The newly hatched nymphs were counted every 24 h, until no new nymphs appeared. The number of unhatched eggs on each plant was counted under a microscope, and the hatching rate and developmental duration of BPH eggs for each treatment were calculated. Each treatment was replicated 9~10 times.

Considering that the developmental duration of BPH eggs in rice tissues laid by BPH female adults is 8–10 days at 25–28 °C [71] and that 1 FAW 3rd-instar larvae will eat all leaves within 5 days (Appendix A), we also assess the impact that the post-infestation of FAW 1st- and 2nd-instar larvae had on the hatching rate and developmental duration of BPH eggs. A total of 10 gravid BPH females were allowed to lay eggs on the leaf sheath of a rice plant for 24 h using the method described above. BPHs were removed, and each plant was infested with two 1st- or three 1st- or 2nd-instar FAW larvae for seven days (controls were kept non-infested). The newly hatched nymphs were counted every 24 h, until no new nymphs appeared. The number of unhatched eggs on each plant was counted under a microscope, and the hatching rate and developmental duration of BPH eggs for each treatment were calculated. Each treatment set had 9~10 biological replicates.

#### 4.3.2. Effects of BPH-Infested Rice Plants on the Performance of FAW Larvae

We first investigated the impact of plants with different damage levels inflicted by gravid BPH females on the growth of freshly hatched FAW larvae. The lower part of the rice plants was infested by 15 gravid BPH females using the method described above (Appendix A). Two or five days later, BPHs were removed or not (glass cages covering plants (Appendix A) were not removed; cages prevent FAW larvae from coming into contact with BPH eggs and eating them), and one newly hatched FAW larva was placed on the upper part of each plant. Plants with empty glass cages (without BPH infestation) were used as controls. FAW larvae were individually weighed 8 or 9 days later. Each treatment was replicated 12~28 times.

Considering that FAW larvae are omnivorous, we assess the impact of plants infested by gravid BPH females, when BPH eggs on these plants can be preyed upon by FAW larvae, on the growth of freshly hatched FAW larvae. The lower part of the rice plants was infested by 15 gravid BPH females using the method described above (Appendix A). Two days later, BPHs, together with glass cages, were removed, and one newly hatched FAW larva was placed on the upper part of each plant. In this experiment, FAW larvae could consume whole plants and reach BPH eggs. Plants without BPH infestation (with empty glass cages) were used as controls. FAW larvae were individually weighed 5.5 days later. Each treatment was replicated 23~30 times. In another experiment, when plants were infested by 15 gravid BPH females for 2 days and BPHs were removed as stated above, these plants were divided into 2 groups. In one group, each plant received one pre-weighed 3rd-instar FAW larva introduced into the glass cage (meaning FAW larvae could reach BPH eggs); in the other group, plants did not receive any FAW larvae. FAW larvae reared on the lower parts of plants without BPH infestation (with empty glass cages) were used as controls. One or three days later, FAW larvae were individually weighed, and the number of eggs on each BPH-infested plant after the release of FAW larvae was counted under a microscope. Each treatment was replicated 20~30 times.

To decide whether the effect of BPH-infested plants on the growth of FAW larvae comes from BPH eggs, we performed the following experiment: pre-weighed 3rd-instar FAW larvae were individually fed on the leaf sheaths of rice plants plus 400 BPH eggs in Petri dishes (diameter 9 cm, height 1.5 cm); FAW larvae fed on rice leaf sheaths alone were used as controls. Rice leaf sheaths and BPH eggs were changed every day, and FAW larvae were weighed 2 and 3 days after the start of the experiment. Each treatment was replicated 28~30 times.

#### 4.3.3. Effect of FAW Larvae Infestation on the Attractiveness of BPH-Infested Plants to the Parasitoid *A. nilaparvatae*

Rice plants were individually infested by 10 gravid BPH females that were confined in a glass cage as described above; meanwhile, one 4th-instar, two 3rd-instar or three 3rd-instar FAW larvae were placed on the rice leaves. Plants infested with BPH alone were used as controls. After 24 h, BPHs and FAW larvae were removed. A BPH-infested plant and a plant co-infested with BPH and FAW were transplanted into a plastic pot (inside diameter 12 cm, height 10 cm) containing 1 L nutrient solution and confined in the plastic cylinder, 4 cm apart (Appendix A, upper inset); 4 pairs of newly emerged parasitic wasps were released into the center of each cylinder. After 48 h, parasitic wasps were removed from the cylinder, and the plants were kept in the climate room. Five days after the release of the parasitoid, the parasitized BPH eggs have turned red. The parasitized and non-parasitized BPH eggs on each plant were counted under a microscope, and the parasitism rate of BPH eggs on plants subjected to different treatments was calculated. Each treatment was replicated 15~19 times.

### 4.4. Analysis of JA, JA-Ile, ABA and SA Levels

Plants were individually infested by 10 gravid BPH females using the method described above. Twenty-four hours later, BPHs were removed, and each plant was infested with three 2nd-instar FAW larvae (BPH + FAW) or kept non-infested (BPH). The outermost part of two leaf sheaths of each plant at one, three, five or seven days after FAW larvae infestation were harvested. Moreover, the same leaf sheaths of plants with empty cages (without BPH and FAW infestation) were also harvested. Samples were ground in liquid nitrogen (about 100 mg per sample for testing), and JA, JA-Ile, SA and ABA were extracted with ethyl acetate containing the labelled internal standards (D6-JA, D6-JA-Ile, D4-SA and D6-ABA) and then isolated and quantified by high-performance liquid chromatography-mass spectrometry-mass spectrometry (HPLC-MS-MS) (Agilent Technologies, Santa Clara, CA, USA) using the method described in [72]. Each treatment was replicated 4~5 times.

### 4.5. Hydrogen Peroxide Analysis

Samples were obtained using the method described above. H_2_O_2_ concentrations were determined using an Amplex^®^ Red Hydrogen Peroxide/Peroxidase Assay Kit (Invitrogen, Eugene, OR, USA) following the instructions. Each treatment was replicated 5 times.

### 4.6. Analysis of Phenolamides and Flavonoids

Samples were obtained using the method described above. Samples were ground in liquid nitrogen (about 100 mg per sample). Phenolamides and flavonoids were extracted with 800 μL of 70% methanol, and then were quantified by HPLC-MS-MS as previously described in [73] and [74], respectively. The contents of each compound were calculated using an external standard method. Each treatment was replicated 5~6 times.

### 4.7. Data Analysis

Two-treatment data were analyzed using Student’s *t* tests. Data from three or more treatments were compared using one-way ANOVA; if the ANOVA analysis was significant (*p* < 0.05), Tukey’s multiple comparison test was used to detect differences between groups. When necessary, data were log- or arc sine-transformed to meet requirements for the homogeneity of variance. All statistical analyses were conducted with SPSS software version 20 (SPSS).

## Figures and Tables

**Figure 1 ijms-24-08754-f001:**
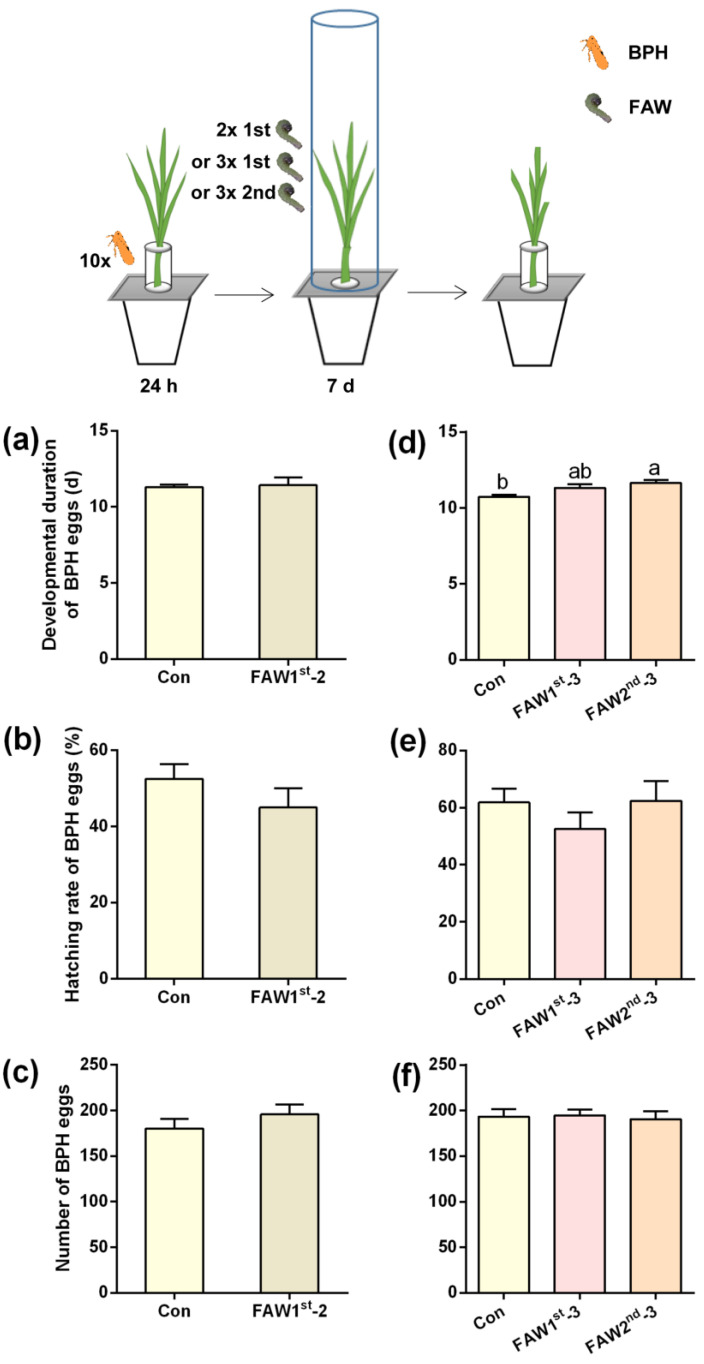
FAW larvae infestation of plants prolongs the developmental duration of BPH eggs. Upper inset, schematic representation of experimental design. (**a**–**f**) Mean developmental duration of BPH eggs (**a**,**d**), hatching rate of BPH eggs (**b**,**e**) and number of eggs laid by 10 BPH gravid female adults for 24 h (**c**,**f**) (+SE, n = 9–10) on non-infested plants (Con) or rice plants that were post-infested by 2 1st-instar (FAW 1st-2), 3 1st^-^instar (FAW 1st-3) or 3 2nd-instar (FAW 2nd-3) FAW larvae for 7 days. Different letters indicate significant differences among treatments in (**d**) (*p* < 0.05, Tukey’s multiple comparison test).

**Figure 2 ijms-24-08754-f002:**
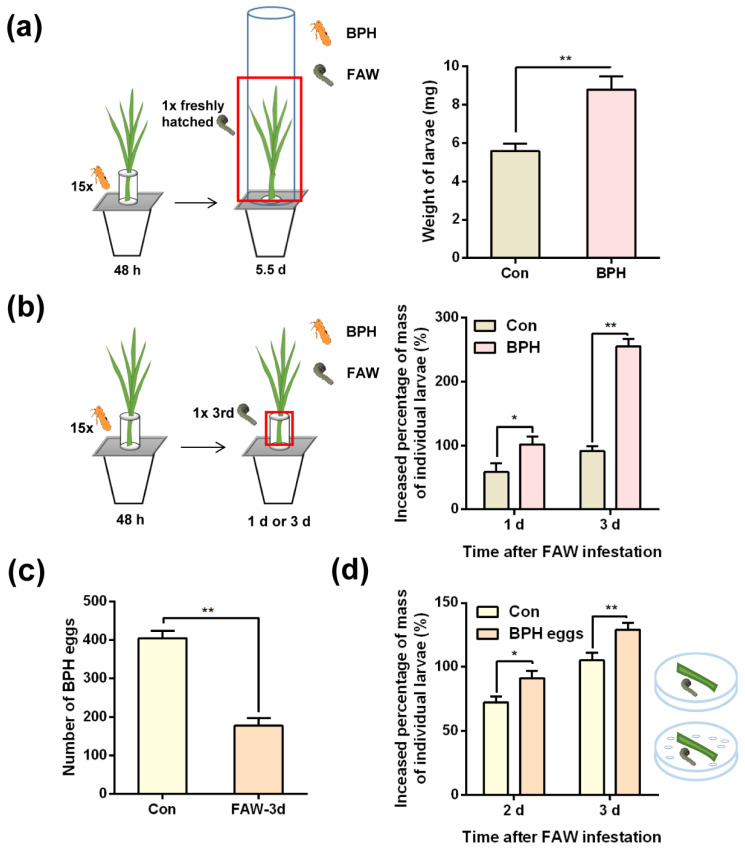
BPH infestation-induced changes in rice do not influence the growth of FAW larvae, but the presence of BPH eggs does. (**a**) Mean mass of individual newly hatched FAW larvae (+SE, n = 23–30) 5.5 days after they fed on whole non-infested plants (Con) or BPH-infested plants (meaning FAW could reach BPH eggs). Inset, schematic representation of the experimental design; the red block indicates the location of FAW larvae placement. (**b**) Mean increased percentage of mass of individual 3rd-instar FAW larvae (+SE, n = 20–22) 1 or 3 days after they fed on lower parts of non-infested plants (Con) or plants that were pre-infested by 15 BPH gravid female adults for 48 h (afterwards, the BPHs were removed). Inset, schematic representation of the experimental design; the red block indicates the location of FAW larvae placement. (**c**) Mean number of remaining BPH eggs (+SE, n = 30) on BPH-infested plants as described (**b**) 3 days after the release of 1 3rd-instar FAW larva (FAW-3d) or not (Con). (**d**) Mean increased percentage of mass of 1 3rd-instar FAW larva (+SE, n = 28–30) 2 or 3 days after it fed on rice leaf sheaths containing BPH eggs or rice leaf sheaths alone (Con). Asterisks indicate significant differences between treatments (* *p* < 0.05; ** *p* < 0.01; Student’s *t* tests).

**Figure 3 ijms-24-08754-f003:**
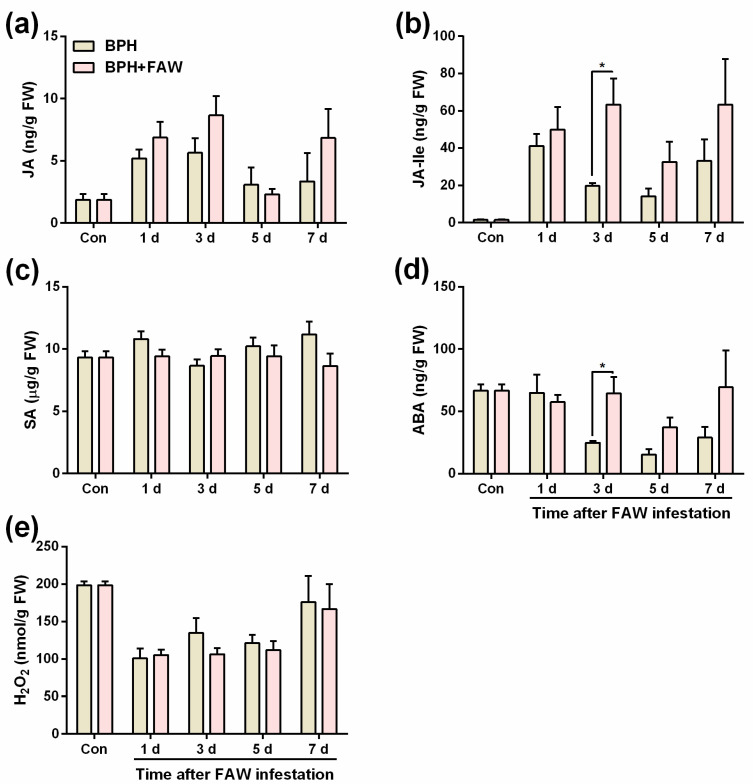
Mean levels (+SE, n = 4~5) of JA (**a**), JA-Ile (**b**), SA (**c**), ABA (**d**) and H_2_O_2_ (**e**) in rice plants that were infested with 10 gravid BPH females for 24 h (and BPHs on them were removed), 1, 3, 5 and 7 days after plants were individually infested with 3 2nd-instar FAW larvae (BPH + FAW) or infested only by BPH. Plants confined in empty cages were used as controls (Con). Asterisks indicate significant difference between treatments (* *p* < 0.05; Student’s *t* tests).

**Figure 4 ijms-24-08754-f004:**
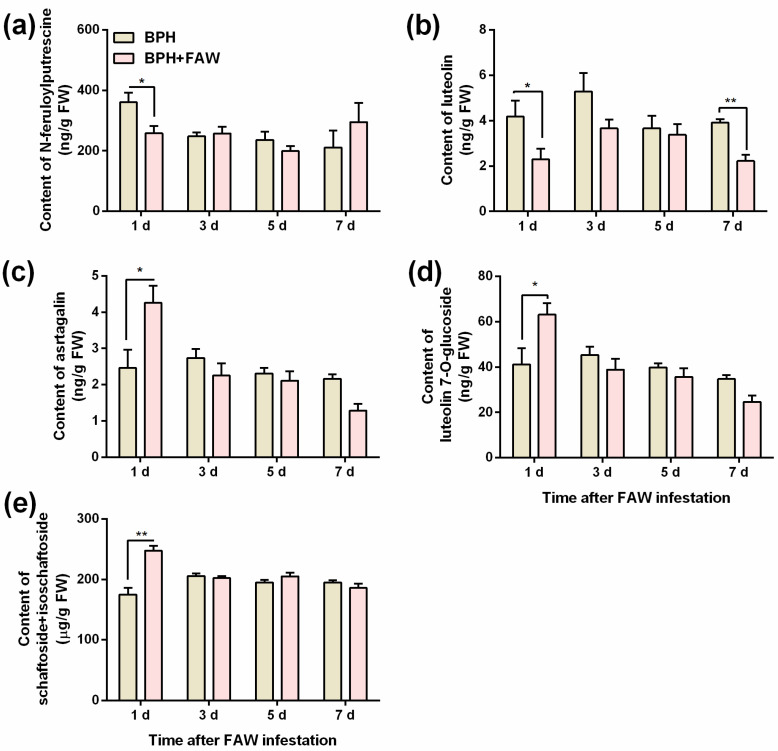
Mean levels (+SE, n = 5~6) of N-feruloylputrescine (**a**), luteolin (**b**), astragalin (**c**), luteolin 7-*O*-glucoside (**d**) and schaftoside + isoschaftoside (**e**) in rice plants that were infested with 10 gravid BPH females for 24 h (and the BPHs on them were removed), 1, 3, 5 and 7 days after plants were individually infested with 3 2nd-instar FAW larvae (BPH + FAW) or controls (infested only by BPH) (BPH). Asterisks indicate significant differences between treatments (* *p* < 0.05, ** *p* < 0.01, Student’s *t* tests).

## Data Availability

The data presented in this study are available on request from the corresponding author.

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
