# Peer review of "The Fall Armyworm, Spodoptera frugiperda (Lepidoptera: Noctuidae), Influences Nilaparvata lugens Population Growth Directly, by Preying on Its Eggs, and Indirectly, by Inducing Defenses in Rice"

_ijms, 2023, doi:10.3390/ijms24108754_

Round 1

Reviewer 1 Report

Add damage estimates for BPH and FAW to highlight more on the importance of these insect pests globally or in China alone.

Provide a description of the sub-objectives after stating the main objective in the introduction section

Include limitation of the study in the discussion section

Line 269: Replace 'feeding on' with 'feeding of'

Line 270-273: Rewrite for clarity

Line 375-376: Rewrite for clarity, facilitate BPH growth or FAW growth?

Add pictures of the the actual experiments, if available

Reviewer 2 Report

In this manuscript, Qiu et al. analyse the interaction between an established rice pest, the brown planthopper (BPH) and a potential invasive rice pest, the fall armyworm (FAW). Interestingly, their findings indicate that if FAW invades rice plants, this species could displace BPH as main pest in rice. These valuable results merit publication. I have a couple of comments/concerns that should be addressed before accepting the manuscript:

  1. In figures 3 and 4 there are not a zero time. This data is important to really know if the amount of hormones or secondary metabolites increase upon BPH infestation.
  2. The methodology used for the quantification of hormones, phenolamides and flavonols should be further described. For example, the reader should know if quantification has been done by HPLC or not, and if authors have modified the original method.

Author Response

Comments and Suggestions for Authors

In this manuscript, Qiu et al. analyse the interaction between an established rice pest, the brown planthopper (BPH) and a potential invasive rice pest, the fall armyworm (FAW). Interestingly, their findings indicate that if FAW invades rice plants, this species could displace BPH as main pest in rice. These valuable results merit publication. I have a couple of comments/concerns that should be addressed before accepting the manuscript:

We thank the reviewer for her/his appreciation for our work.

1.In figures 3 and 4 there are not a zero time. This data is important to really know if the amount of hormones or secondary metabolites increase upon BPH infestation.

Response: We thank the reviewer for pointing these out. We have added the phytohormone data at zero time point. For the secondary metabolites, we agree with the reviewer’s idea that if we measured the level of these compounds at zero time, we will know whether the production of these compounds was induced by BPH infestation. However, in this study, we mainly want to explain the reason why FAW larvae infestation delayed the developmental duration of BPH eggs. Thus, we think the most important thing is that whether the levels of these defensive compounds in BPH+FAW-infested plants is higher than those in BPH-infested plants during the developmental process of BPH eggs (at 1-7 d after FAW larvae infestation)(not at zero time). Therefore, we have not added the data of these compounds at zero time point.

2.The methodology used for the quantification of hormones, phenolamides and flavonols should be further described. For example, the reader should know if quantification has been done by HPLC or not, and if authors have modified the original method.

Response: We thank the reviewer for these comments. We have improved the text accordingly.